# Discovery of novel TACE inhibitors using graph convolutional network, molecular docking, molecular dynamics simulation, and Biological evaluation

**Muhammad Yasir**[1]☯, **Jinyoung Park**[1]☯, **Eun-Taek Han**[2], **Jin-Hee Han**[2], **Won Sun Park**[3], **Mubashir Hassan**[4], **Andrzej Kloczkowski**[4], **Wanjoo Chun**[1]*

1 Department of Pharmacology, Kangwon National University School of Medicine, Chuncheon, Republic of Korea, 2 Department of Medical Environmental Biology and Tropical Medicine, Kangwon National University School of Medicine, Chuncheon, Republic of Korea, 3 Department of Physiology, Kangwon National University School of Medicine, Chuncheon, Republic of Korea, 4 The Steve and Cindy Rasmussen Institute for Genomic Medicine at Nationwide Children's Hospital, Columbus, Ohio, United States of America

☯ These authors contributed equally to this work.
* wchun@kangwon.ac.kr

**Data Availability Statement:** All relevant data are within the paper, Supporting information file and

## Abstract

The increasing utilization of deep learning models in drug repositioning has proven to be highly efficient and effective. In this study, we employed an integrated deep-learning model followed by traditional drug screening approach to screen a library of FDA-approved drugs, aiming to identify novel inhibitors targeting the TNF-α converting enzyme (TACE). TACE, also known as ADAM17, plays a crucial role in the inflammatory response by converting pro-TNF-α to its active soluble form and cleaving other inflammatory mediators, making it a promising target for therapeutic intervention in diseases such as rheumatoid arthritis. Reference datasets containing active and decoy compounds specific to TACE were obtained from the DUD-E database. Using RDKit, a cheminformatics toolkit, we extracted molecular features from these compounds. We applied the GraphConvMol model within the DeepChem framework, which utilizes graph convolutional networks, to build a predictive model based on the DUD-E datasets. Our trained model was subsequently used to predict the TACE inhibitory potential of FDA-approved drugs. From these predictions, Vorinostat was identified as a potential TACE inhibitor. Moreover, molecular docking and molecular dynamics simulation were conducted to validate these findings, using BMS-561392 as a reference TACE inhibitor. Vorinostat, originally an FDA-approved drug for cancer treatment, exhibited strong binding interactions with key TACE residues, suggesting its repurposing potential. Biological evaluation with RAW 264.7 cell confirmed the computational results, demonstrating that Vorinostat exhibited comparable inhibitory activity against TACE. In conclusion, our study highlights the capability of deep learning models to enhance virtual screening efforts in drug discovery, efficiently identifying potential candidates for specific targets such as TACE. Vorinostat, as a newly identified TACE inhibitor, holds promise for further

online on GitHub (https://github.com/Yasirubunt/TACE_FDA.git)

**Funding:** This work was supported by a Research Grant from the Institute of Medical Sciences, Kangwon National University 2024, a Korea Basic Science Institute (National Research Facilities and Equipment Center) grant funded by the Ministry of Education (grant no. 2022R1A6C101A739), National Institute of Health grant numbers R01GM127701 and R01HG012117. The funders had no role in study design, data collection and analysis, decision to publish, or preparation of the manuscript.

**Competing interests:** The authors have declared that no competing interests exist.

exploration and investigation in the treatment of inflammatory diseases like rheumatoid arthritis.

## Introduction

Tumor necrosis factor-alpha (TNF-α) plays a pivotal role in the inflammatory response and is a key cytokine involved in the pathogenesis of various inflammatory diseases, including rheumatoid arthritis [1,2]. The biological activities of TNF-α extend beyond inflammation to include the regulation of immune cells, apoptosis, and inhibition of tumorigenesis, highlighting its dual role in health and disease [3–6]. Understanding the regulation and function of TNF-α is essential for developing therapeutic strategies aimed at controlling inflammatory conditions. TNF-α is synthesized as a 26 kDa transmembrane precursor, known as pro-TNF-α. This precursor form undergoes a proteolytic cleavage to produce a 17 kDa soluble mature protein, a process catalyzed by the TNF-α converting enzyme (TACE) [7,8]. TACE, also known as a disintegrin and metalloproteinase 17 (ADAM17), is a membrane-bound enzyme that plays a crucial role in the conversion of pro-TNF-α to its active soluble form [9–11].

The activity of TACE is not limited to TNF-α; it also cleaves a variety of other cell surface proteins. The broad substrate specificity of TACE underscores its significant role in modulating various physiological and pathological processes. By cleaving cytokines and their receptors, TACE influences cytokine signaling pathways, which are crucial for immune responses and inflammation. For instance, the cleavage of IL-6R by TACE leads to the activation of IL-6R-mediated signal transduction, contributing to the inflammatory cascade [12,13]. Similarly, TACE-mediated cleavage of TNF-R influences TNFR-mediated signaling pathways, which are vital for immune regulation and inflammatory responses [14,15]. The modulation of these pathways by TACE suggests that its activity is a critical pro-inflammatory event. The role of TACE in inflammation is further exemplified by its involvement in the shedding of ligands of the ErbB receptor family. ErbB receptors are implicated in cell growth, differentiation, and survival, and their ligands, such as TGFα and amphiregulin, are essential for the activation of these receptors. TACE-mediated cleavage of these ligands affects ErbB signaling pathways, which have been associated with inflammation and tissue remodeling [16]. Additionally, the cleavage of adhesion molecules like ICAM-1 by TACE modulates leukocyte adhesion and migration, processes that are fundamental to the inflammatory response [17]. Furthermore, TNF-α is also produced by neurons and glia, plays a significant role by promoting inflammatory responses and possibly inducing Aβ production in neuroinflammations. TACE is crucial in this process, converting precursor TNF-α to its soluble form and cleaving TNFR1 and TNFR2 to their soluble counterparts. Elevated levels of TNF-α, sTNFR1, and sTNFR2 in the cerebrospinal fluid and plasma are associated with Alzheimer's disease (AD) and mild cognitive impairment (MCI), and higher TACE activity has been observed in both MCI and AD patients. This underscores the importance of TNF-α and TACE in the inflammatory pathology of AD [18,19].

Given the central role of TACE in converting pro-TNF-α to its soluble form and its involvement in the regulation of other inflammatory mediators, TACE has emerged as a potential therapeutic target for inflammatory diseases [18,20]. The inhibition of TACE activity could potentially reduce the levels of soluble TNF-α and other pro-inflammatory molecules, thereby mitigating inflammation and its associated pathologies. Several studies have focused on developing TACE inhibitors as therapeutic agents for conditions such as rheumatoid arthritis,

psoriasis, and other inflammatory disorders [9,21–23]. These inhibitors aim to block the proteolytic activity of TACE, preventing the release of soluble TNF-α and other substrates, and consequently dampening the inflammatory response.

Recently, the integration of deep learning models in the drug discovery process has gained significant attention therefore, this study explores its potential for the identification of highly efficient and novel TACE inhibitors. These models leverage large datasets and algorithms to screen the compounds with potential binding affinities and activities, thus streamlining the discovery of effective therapeutic candidates. By incorporating deep learning into traditional drug screening, researchers can rapidly screen vast libraries of compounds to identify promising drug candidates with high reliability, leading to further biological validations and trials therefore, fasten the drug discovery process.

## Materials and methods

### TACE dataset, FDA-approved drug library, and commercial Enamine compound library

The DUD-E website provided access to datasets containing TACE active and decoy compounds (https://dude.docking.org/) (accessed on 01 May 2024). The active dataset included 532 compounds, while the decoy dataset comprised 35,900 compounds. Each molecule was represented by canonical SMILES strings, accompanied by corresponding DUD-E ID and ChEMBL ID numbers, with the legend distinguishing between active and decoy compounds. Additionally, an FDA-approved drug library consisting of 3,105 compounds was obtained from the Selleck Chemicals website (https://www.selleckchem.com). Initially formatted in SDF (structure-data file), the FDA-approved drugs were converted into SMILES strings using RDKit. Furthermore, BMS-561392, a known TACE inhibitor, was obtained from MedChemExpress (https://www.medchemexpress.com/) in SMILES format, serving as a reference compound.

### Molecular descriptor generation using RDKit

Molecular descriptors for all the compounds were generated using RDKit, an open-source, high-performance cheminformatics and machine learning toolkit available at https://www.rdkit.org. This toolkit was implemented in Python and provides functionalities for computing molecular descriptors, creating chemical features, and visualizing chemical data.

### Deep learning architecture

The TACE active and decoy datasets were split into training, validation, and test sets using an 8:1:1 ratio. Following this division, we conducted deep learning evaluations using the GraphConvModel from DeepChem (https://deepchem.io/models). GraphConvModel, a type of graph convolutional neural network, is particularly adept at learning features from graph-structured input data such as molecular graphs. This model preprocesses molecular structures into graphs, where atoms represent nodes and bonds represent edges. Node features are extracted using the ConvMolFeaturizer, which generates atom-level features such as atom type, formal charge, chirality, degree, number of hydrogens, and hybridization state. Edge features correspond to the adjacency list of the molecule, indicating the connectivity between atoms.

The model includes two graph convolutional layers with 64 hidden units each, followed by batch normalization and ReLU activation to stabilize and accelerate training while introducing non-linearity. A dense layer with 128 units is added after the graph convolutional layers, with a

dropout rate of 0.25 to prevent overfitting. The readout layer aggregates node features to produce a graph-level representation, and the final output layer is configured for binary classification with a single output. The model uses a batch size of 128 and saves the trained model. For the dataset, the RandomSplitter is employed to split the dataset, and the model's performance is evaluated using the Matthews correlation coefficient (MCC) as the primary metric.

Throughout the training process, the model aims to minimize the loss function by analyzing the input molecular datasets. Using backpropagation, it iteratively adjusts the weights of the convolutional layers to improve predictions of properties such as solubility, bioactivity, and toxicity based on the given molecular structures [24]. This architecture effectively captures the relational information between atoms in molecules, enabling accurate predictions for molecular property prediction and drug discovery tasks. Detailed information on the model's architecture was provided as a Colab Jupyter Notebook file on a GitHub repository (https://github.com/Yasirubunt/yasirknu.github.io).

## Accession of TACE crystalized structure

The three-dimensional structure of the TACE, identified by the PDB ID: 2OI0 with a resolution of 2.00Å, was acquired from the Protein Data Bank (PDB) accessible at https://www.rcsb.org/. The TACE crystal structure, comprising alpha-helices, beta-sheets, coils, and turns, underwent a comprehensive quantitative structural analysis utilizing the VADAR internet server (http://vadar.wishartlab.com/). Subsequently, energy minimization and Ramachandran Plot analysis were carried out utilizing UCSF Chimera and Discovery Studio [25,26].

## Prediction of active binding site

The positioning of a ligand within a protein's holo-structure is a key determinant of the protein's binding pocket [27]. The TACE and co-crystallized ligand, already available on PDB (PDB ID: 2OI0), was further utilized for binding pocket analysis. The identification of interacting amino acids was accomplished through Discovery Studio's ligand interaction approach, ensuring precision in the generation of the binding site. Additionally, the binding pocket residues were confirmed from already published data [28]. Therefore, the bound ligand was chosen by ligand selection, and the binding sphere was created using the current selection approach within the defined Binding Site window of Discovery Studio. Subsequently, the binding sphere was reduced with imposed restrictions specific to the selected amino acids for the accuracy and precision of docking.

## Molecular docking

Molecular docking stands as the extensively employed approach for assessing the interactions between ligands and receptors [29]. It forecasts the strength of association or binding energy of protein-ligand complexes by assessing their preferred orientations, using scoring algorithms in the process [24]. The already bound ligand molecule and water molecules were deleted from the protein. TACE is a metalloproteinase therefore $Zn2+$ ion was retained in its native position to accurately represent the binding environment, as it plays a crucial role in ligand binding. The coordination geometry and charge state of the $Zn2+$ ion were carefully maintained to ensure realistic interactions with the ligands.

Furthermore, the hydrogens were appended to the protein utilizing Discovery Studio's receptor preparation module. Moreover, the ligand preparations for both reference and candidate compounds included the generation of tautomers, adjustment of ionization states, and correction of any bad valences. These tasks were performed using the Ligands Preparation module in Discovery Studio. The molecular docking of ligands against the target protein

(TACE) was conducted using the CDocker module in Discovery Studio, employing default orientation and conformation settings. The presence of the Zn2+ ion was reconfirmed before starting the molecular docking, ensuring that the ligands could form interactions with it where applicable. Therefore, the docked complexes were assessed based on the docking energy score kcal/mol.

## Molecular dynamics simulations

The Molecular Dynamics (MD) simulation protocols followed were adapted from our previously published paper involving a 100 ns simulation [30], in which we screened JAK2 inhibitors. Therefore, the top seven compounds with the lowest docking energies were selected for the MD simulation, alongside BMS-561392 as a reference compound for comparative analysis. The CHARMM-GUI web server (https://www.charmm-gui.org/?doc=input/solution) was used to generate input files for MD simulations with GROMACS [31]. For each complex, TACE, the ligand, and the $Zn^{2+}$ ion were submitted to the CHARMM-GUI web server, where the solution builder protocol was employed to set up the system using the CHARMM36 force field [32]. The system was solvated using the TIP3P in a cubic box under periodic boundary conditions [33], with neutralization achieved by adding counter ions. Electrostatic and van der Waals interactions were computed using the Verlet method with a cutoff radius of 10, and bond lengths were maintained using the LINCS algorithm during simulations. The Particle Mesh Ewald (PME) method was employed for accurate long-range electrostatic calculations, providing precision in the modeling of interactions involving the Zn2+ ion and surrounding residues.

Energy minimization using the steepest descent method prepared the solvated systems. Two equilibration phases followed, first under constant temperature and volume (NVT) conditions and then under constant temperature and pressure (NPT) conditions. Conversion of GROMACS topology (top) and parameter (itp) files for MD simulations was facilitated by a Python script from CHARMM-GUI, ensuring consistency and accuracy of the force field parameters, particularly for the Zn2+ coordination environment. In the end, structural analysis of protein-ligand complexes was conducted using GROMACS version 2019.3 on a Linux system [34], with a time step of 2fs used for the MD simulations.

## Binding free energy calculation of simulated compound

Predictions of binding free energy were made using an MM/PBSA approach from MD simulation trajectories in explicit solvent, examining the three components complex, receptor, and ligand individually [35]. The binding free energy (ΔGbinding) of the lead compounds in complex with the protein was calculated using the following equation:

$$\Delta Gbinding = Gcomplex - (Gprotein + Gligand) \tag{1}$$

In this equation, Gcomplex denotes the energy of the lead compounds-protein complexes, while Gprotein and Gligand represent the energies of the proteins and ligands in an aqueous environment, respectively [47].

## Reagents and cell culture

Bacterial lipopolysaccharide (LPS) from Escherichia coli serotype 055:B5 was purchased from Sigma-Aldrich (St. Louis, MO, USA). BMS-561392, Quinapril hydrochloride, Marimastat, and Bufexamac, were purchased from MedChemExpress (Monmouth Junction, NJ, USA). Vorinostat and Thiamine hydrochloride were purchased from Sigma-Aldrich (St. Louis, MO, USA).

Raw 264.7 macrophage cells were purchased from the Korea cell line bank (KCLB, cat #40071). Raw 264.7 macrophage cells were maintained in Dulbecco's modified Eagle's medium (DMEM; BYLABS, Luscience Corporation, Korea) containing 10% heat-inactivated fetal bovine serum and 100 U/ml penicillin-streptomycin (Gibco) at 37 ˚C, 5% CO2. Cells were incubated in 10 μM of various reagents for 1h and then stimulated with 1 μg/ml of LPS for 4h.

### ELISA assay for cytokine

Raw 264.7 macrophage cells were pretreated with various compounds (10 μM) for 1h and then stimulated with 1 μg/ml of LPS for 4h. TNF-α levels in culture media were quantified using enzyme-linked immunosorbent (ELISA) kits (R&D system, USA) according to the manufacturer's instructions.

### Statistical analysis

All values presented in Fig 10 were expressed as the mean ± SD, derived from a minimum of three independent experiments. Statistical significance was assessed using a two-tailed Student's t-test, with a value of $p<0.05$ considered statistically significant. The double (**) asterisk denotes statistical significance at $p < 0.01$.

## Results and discussion

The discovery process of novel TACE inhibitors integrates ligand-based virtual screening with a deep learning algorithm and structure-based virtual screening using molecular docking and molecular dynamics (MD) simulations, illustrated in Fig 1. This study proceeded through six distinct stages: First, datasets were acquired from the DUD-E database. Second, target-specific datasets were prepared for the deep learning model. Third, the GraphConvMol algorithm was configured. Fourth, TACE inhibitory activity was predicted using the trained GraphConvMol model. Fifth, candidates underwent screening via molecular docking and MD simulations. Finally, experimental validation of the candidates was conducted using an enzyme activity assay.

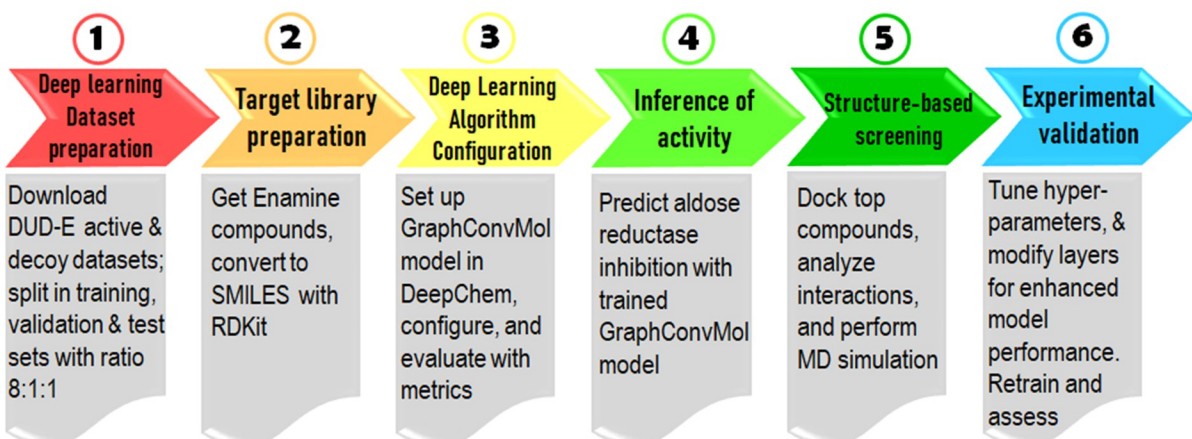

**Fig 1. The workflow combines deep learning, molecular docking, MD simulations, and experimental evaluation to identify new TACE inhibitors from the pool of FDA-approved drugs.**

## TACE active and decoy datasets and its preprocessing using RDKit

DUD-E (Database of Useful Decoys: Enhanced) provides benchmark sets of protein-ligand complexes, featuring experimentally verified active compounds alongside decoys that lack binding affinity against various diseases. These decoys exhibit comparable physicochemical characteristics to the active compounds, yet they diverge in their two-dimensional topology [36]. The DUD-E database is extensively utilized for developing and benchmarking computational docking methods [37,38].

The TACE dataset in the DUD-E repository (https://dude.docking.org/targets/ADA17) consists of 532 experimentally validated active compounds. Fig 2 illustrates representative structures of active and decoy compounds, with labels identifying their respective classifications. Molecular features, including molecular weight, LogP, Hbond donors, Hbond acceptors, TPSA, and rotatable bonds, were computed using RDKit, to facilitate comparison of the physicochemical properties between active and decoy compounds. No significant variance was observed in the distribution of molecular descriptor values between active and decoy compounds (Fig 3).

## Deep-learning model setup, training, and evaluation

DeepChem is an open-source Python library designed for applying deep learning to drug discovery and cheminformatics. This versatile library provides a comprehensive set of tools for managing molecular data. DeepChem effectively utilizes a variety of deep-learning algorithms to address tasks in drug discovery, such as predicting molecular properties, performing ligand-based virtual screening, and optimizing chemical compounds [39,40].

In this study, the GraphConvMol model from the DeepChem library played a crucial role in distinguishing between active and decoy compounds in the TACE dataset. The GraphConvMol method comprehensively learns molecular representations, making it a powerful tool for cheminformatics applications such as predicting molecular properties and aiding in drug

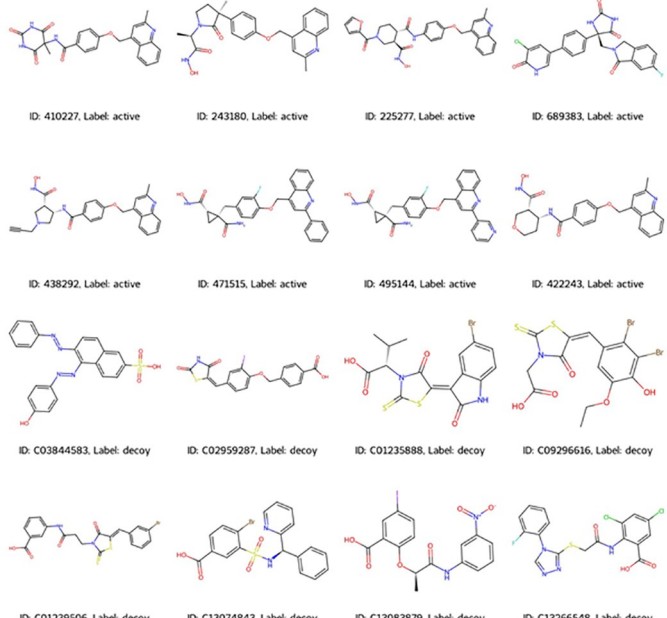

**Fig 2. Illustrative depiction of active and decoy compounds.**

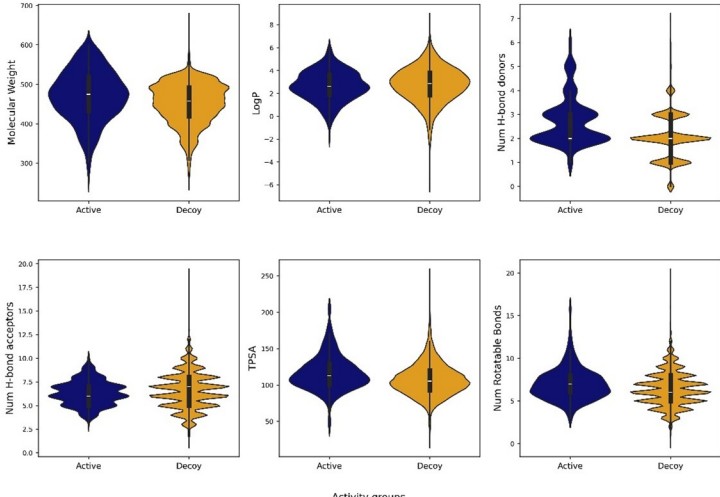

**Fig 3. Distribution of molecular descriptors in active and decoy compounds.**

discovery [41,42]. The TACE dataset was divided into training, validation, and test sets in an 8:1:1 ratio. The GraphConvMol model was then applied with 5-fold cross-validation.

To evaluate the model's performance, the Area Under the Curve (AUC) of the Receiver Operating Characteristic (ROC) curve was calculated for the validation and test datasets, with the training dataset included for monitoring model fitting. The ROC curve from the 5-fold cross-validation on the training dataset demonstrated near-perfect performance, with a high Area Under the Curve (AUC) value of 0.99574, indicating a True Positive Rate (TPR) close to 1 at very low False Positive Rates (FPR) (Fig 4). These results highlight the GraphConvMol model's exceptional sensitivity in accurately identifying positive instances.

In the DUD-E dataset for TACE, the GraphConvMol model achieved exceptional performance, with all evaluated metrics precision, recall, F1 score, sensitivity, accuracy, and specificity—reaching the maximum possible value of 1.0. These results indicate that the model

**A** **B**

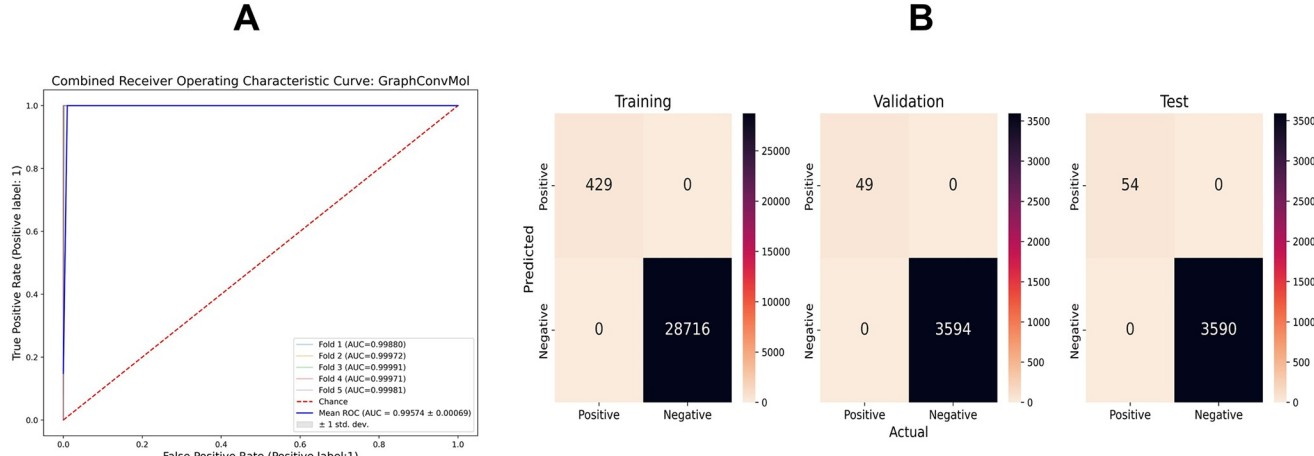

**Fig 4. The AUC-ROC curve of 5-fold cross-validation of the training dataset (A) and the confusion matrix entries for the training, validation, and test datasets (B).**

accurately classified all compounds as either TACE inhibitors or non-inhibitors without any errors, highlighting its effectiveness.

Given the imbalance in the current dataset, with significantly more decoys than active molecules, the Matthews correlation coefficient (MCC) was used to evaluate the GraphConvMol model's efficacy due to its suitability for unbalanced datasets. The average MCC values from five-fold cross-validations were 1.00 for both the training and validation sets. An MCC of 1 indicates perfect prediction accuracy, highlighting the robustness and reliability of the model.

### Prediction of TACE inhibitory potential from FDA-approved drug compound libraries

Using FDA-approved drugs to discover novel TACE inhibitors offers several advantages. Since these drugs have already undergone extensive pre-clinical and clinical testing to establish their safety, dosing, and pharmacokinetics, repurposing them can lead to faster development timelines, reduced costs, and a higher likelihood of success. In this study, an FDA-approved drug library with a unique collection of 3104 approved drugs and Active Pharmaceutical Ingredient (API) included in pharmacopeia for high throughput screening (HTS) and high content screening (HCS), was utilized, available at https://www.selleckchem.com/screening/fda-approved-drug-library.html. Therefore, the SMILES strings of FDA-approved drugs were analyzed using the trained GraphConvMol model from DeepChem to predict their potential TACE inhibitory activity. The trained model provided predictions for the compounds, ranging from 0 (indicating no activity) to 1 (indicating high activity). Most compounds were predicted to be inactive, with only a small proportion predicted as active. Out of 3,104 FDA-approved drugs, 28 compounds were predicted to be active, with prediction values exceeding 0.90 (Table 1). For the next phase of structure-based virtual screening, we chose the top FDA-approved drugs that received high predictions from the GraphConvMol model. Additionally, we included BMS-561392, a confirmed TACE inhibitor available from MedChemExpress LLC (https://www.medchemexpress.com). These compounds were selected for molecular docking.

### Structural analysis of the TACE protein

TACE consists of a single chain of 266 amino acids, comprising α-helices, β-sheets, and coils (Fig 5). According to VADAR statistical analysis, TACE's composition includes 30% α-helices, 27% β-sheets, 42% coils, and 22% turns. Moreover, the Ramachandran plot analysis indicated that 96.2% of all residues fall within favored regions, 100% are in allowed regions, and there are no outliers for the dihedral angles phi ($\varphi$) and psi ($\psi$).

### The binding pocket analysis

The binding pocket is a structurally defined site on a protein, determined by its shape, location, and the assembly of amino acid residues that dictate its functionality [43]. The binding pocket residues of TACE were obtained using Discovery Studio's ligand interaction approach in which the 3D structure of TACE and co-crystallized ligand (PBD ID: 2OI0) was utilized to figure out the interacting amino acid and mentioned as Thr347, Leu348, Leu401, His405, Glu406, His409, Ala439, His415, Val434, Tyr436, and Pro437. Consequently, the verification of active pocket residues was done by comparing them with already published data [28]. Therefore, the binding sphere values were adjusted as X = 44.4357, Y = -28.4975, Z = 3.0389, and the radius value was fixed as 8.1601 concerning the binding pocket residues, to carry out the molecular docking of screened FDA drugs against the active region of target protein (Fig 6).

**Table 1. Top screened compounds from the pool of FDA library.**

| SMILES | Neg | Pos | Name | Target/action |
|---|---|---|---|---|
| C[C@H](N[C@@H](CCc1ccccc1)C (= O)O)C (= O)N1CCC [C@H]1C (= O)O.O.O | 0.0009 | 0.9991 | Enalaprilat Dihydrate | ACE inhibitor |
| Cl.N[C@@H](Cc1cnc[nH]1)C (= O)O.O | 0.0011 | 0.9989 | L-Histidine Hydrochloride Hydrate | Endogenous metabolite |
| Cl.N = C1CCCN1Cc1[nH]c (= O)[nH]c (= O)c1Cl | 0.0025 | 0.9975 | Tipiracil Hydrochloride | Thymidine phosphorylase inhibitor (TPI) |
| O = C(CCCCCCC (= O)Nc1ccccc1)NO | 0.0047 | 0.9953 | Vorinostat | Histone deacetylase inhibitor |
| CCCCOc1ccc(CC (= O)NO)cc1 | 0.0048 | 0.9952 | Bufexamac | Anti-inflammatory, analgesic, antipyretic |
| Cc1ncc(C[n+]2csc(CCO)c2C)c(N)n1.Cl.[Cl-] | 0.0050 | 0.9950 | Thiamine Hydrochloride | Vitamin B1 |
| CC (= O)[O-].CC (= O)[O-].O.O.[Zn+2] | 0.0083 | 0.9917 | Zinc Acetate Dihydrate | Nutritional supplement |
| N.N.O = C([O-])C1(C (= O)[O-])CCC1.[Pt+2] | 0.0102 | 0.9898 | Paraplatin | Antineoplastic |
| Cl.Cl.Cl.Cl.Nc1ccc(-c2ccc(N)c(N)c2)cc1N.O | 0.0106 | 0.9894 | 3,3'-Diaminobenzidine Tetrahydrochloride Hydrate | |
| O = C1N = C([O-])NC1(c1ccccc1)c1ccccc1.[Na+] | 0.0107 | 0.9893 | Phenytoin | Anticonvulsant |
| Cl.c1ccc2c(CC3 = NCCN3)cccc2c1 | 0.0152 | 0.9849 | Naphazoline Hydrochloride | Alpha-adrenergic agonist |
| Cc1nc[nH]c1CN1CCc2c(c3ccccc3n2C)C1 = O.Cl | 0.0165 | 0.9835 | Alosetron Hydrochloride | 5-HT3 receptor antagonist |
| CNC (= O)[C@@H](NC (= O)[C@H](CC(C)C)[C@H](O)C (= O)NO)C(C)(C)C | 0.0219 | 0.9782 | Marimastat | Matrix metalloproteinase inhibitor |
| Cc1ncc(C[n+]2csc(CCO)c2C)c(N)n1.[Cl-] | 0.0313 | 0.9687 | Vitamin B1 | Vitamin B1 |
| C = C[C@H]1CN2CC[C@H]1C[C@H]2[C@H](O)c1ccnc2ccc(OC)cc12.Cl.O.O | 0.0382 | 0.9618 | Quinine Hydrochloride Dihydrate | Antimalarial |
| Cl.N = C(N)NC (= N)NCCc1ccccc1 | 0.0387 | 0.9613 | Phenformin | Anti-diabetic |
| Cc1cccc(Cc2c[nH]cn2)c1C.Cl | 0.0415 | 0.9585 | Detomidine Hydrochloride | Alpha-adrenergic agonist |
| Cc1nccn1CC1CCc2c(c3ccccc3n2C)C1 = O.Cl.O.O | 0.0451 | 0.9549 | Ondansetron Hydrochloride Dihydrate | Antiemetic, 5-HT3 receptor antagonist |
| CCc1ccc(Cc2ccc3c(c2)[C@]2(OC3)O[C@H](CO)[C@@H](O)[C@H](O)[C@H]2O)cc1.O | 0.0460 | 0.9540 | Tofogliflozin Hydrate | SGLT2 inhibitor |
| CCN1CC(CCN2CCOCC2)C(c2ccccc2)(c2ccccc2)C1 = O.Cl.O | 0.0486 | 0.9515 | Doxapram Hydrochloride Monohydrate | CNS stimulant |
| C[C@H](O)C (= O)[O-].[Na+] | 0.0544 | 0.9456 | Sodium L-Lactate | Antimicrobial |
| Cl.c1ccc(CC2 = NCCN2)cc1 | 0.0552 | 0.9449 | Tolazoline Hydrochloride | Alpha-adrenergic antagonist |
| Cl.N = C(N)c1ccccc1 | 0.0659 | 0.9341 | Benzamidine Hydrochloride | Serine protease inhibitor |
| Cl.Cl.c1cnc2cc3c(cc2n1)C1CNCC3C1 | 0.0798 | 0.9202 | Varenicline Dihydrochloride | Nicotinic acetylcholine receptor agonist |
| CCCN[C@H]1CCc2nc(N)sc2C1.Cl.Cl.O | 0.0804 | 0.9196 | Pramipexole Dihydrochloride Monohydrate | Antiparkinsonian |
| CC (= O)[O-].[Na+] | 0.0812 | 0.9188 | Sodium Acetate | Buffering agent |
| CC(C (= O)[O-])c1ccc(CC2CCCC2 = O)cc1.[Na+] | 0.0818 | 0.9182 | Loxoprofen Sodium | Anti-inflammatory, analgesic, antipyretic |
| Cc1cc(C(C)(C)C)cc(C)c1CC1 = NCCN1.Cl | 0.0887 | 0.9113 | Xylometazoline Hydrochloride | H1 histamine receptor antagonist |
| Cc1nc(C)c(C)nc1C.Cl | 0.1108 | 0.8892 | Ligustrazine Hydrochloride | Antioxidant, anti-inflammatory |
| Cl.c1ccc(Nc2ccccc2)cc1 | 0.1120 | 0.8881 | Diphenylamine Hydrochloride | H1 histamine receptor antagonist |
| Cc1ccc(/C (= C\CN2CCCC2)c2ccccn2)cc1.Cl.O | 0.1178 | 0.8822 | Triprolidine HCL | H1 histamine receptor antagonist |
| C[C@H]1CN(C[C@H](Cc2ccccc2)C (= O)NCC (= O)O)CC[C@@]1(C)c1cccc(O)c1.O.O | 0.1203 | 0.8797 | Alvimopan Dihydrate | μ-Opioid receptor antagonist |
| Cl.N = C(N)NC (= N)N1CCOCC1 | 0.1289 | 0.8711 | Moroxydine Hydrochloride | Antiviral |
| O = C(CS (= O)C(c1ccccc1)c1ccccc1)NO | 0.1298 | 0.8702 | Adrafinil | Alpha-adrenergic agonist |
| COc1ccc2cc([C@H](C)C (= O)[O-])ccc2c1.[Na+] | 0.1584 | 0.8416 | Naproxen Sodium | Cyclooxygenase inhibitor |
| Cc1ccc(S (= O) (= O)O)cc1.O | 0.1611 | 0.8389 | p-Toluenesulfonic Acid Monohydrate | Antimalarial |
| Cl.Cl.O = C(O)COCCN1CCN(C(c2ccccc2)c2ccc(Cl)cc2)CC1 | 0.1801 | 0.8199 | Cetirizine Dihydrochloride | Antihistaminic |
| CCOC (= O)[C@H](CCc1ccccc1)N[C@@H](C)C (= O)N1Cc2ccccc2C[C@H]1C (= O)O.Cl | 0.1854 | 0.8146 | Quinapril Hydrochloride | Ace inhibitor |
| Cl.Cl.Clc1cccc(C(c2ccc3nc[nH]c3c2)n2ccnc2)c1 | 0.1954 | 0.8046 | Liarozole Dihydrochloride | Retinoic Acid (Ra) metabolism-blocking agent (Ramba) |
| Cl.N = C(N)N | 0.2027 | 0.7973 | Guanidine Hydrochloride | Protein denaturant |

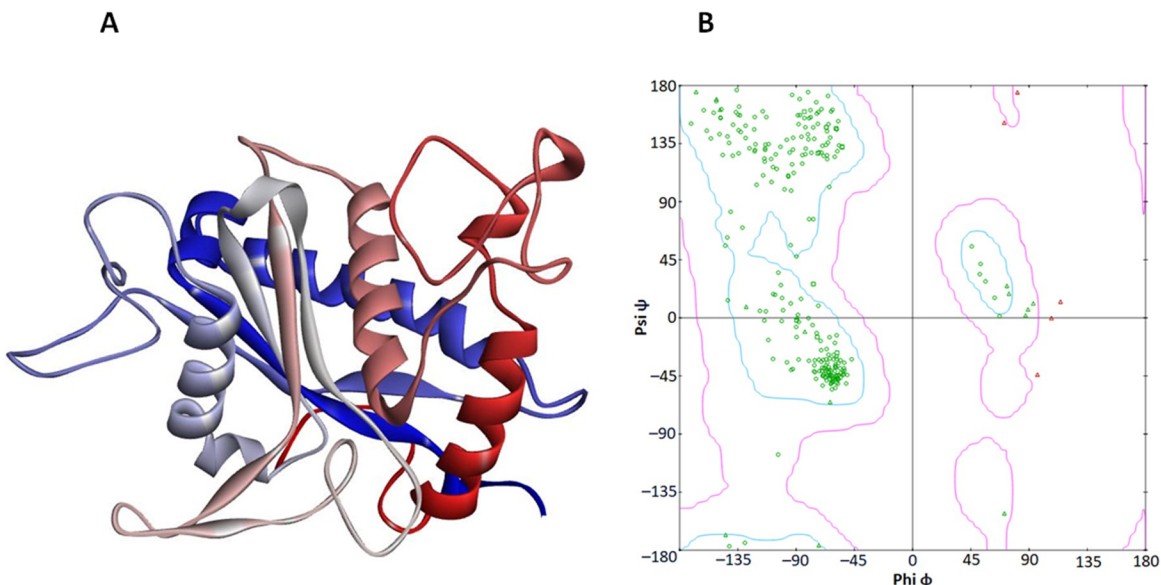

**Fig 5. N-terminal to C-terminal 3D structure (A) of the TACE enzyme and the computed Ramachandran plot (B) showing 0 outliers.**

## Molecular docking analysis

The CDocker module of Discovery Studio was employed for molecular docking. CDocker energy represents the overall docking energy calculated from the 3D structural and physico-chemical characteristics of the ligand and protein complex. On the other hand, CDocker inter-action energy specifically quantifies the energy associated with interactions, including hydrogen bonding, van der Waals forces, and electrostatic interactions [9]. From the subset of top screened compounds, 33 compounds including screened FDA drugs and a reference

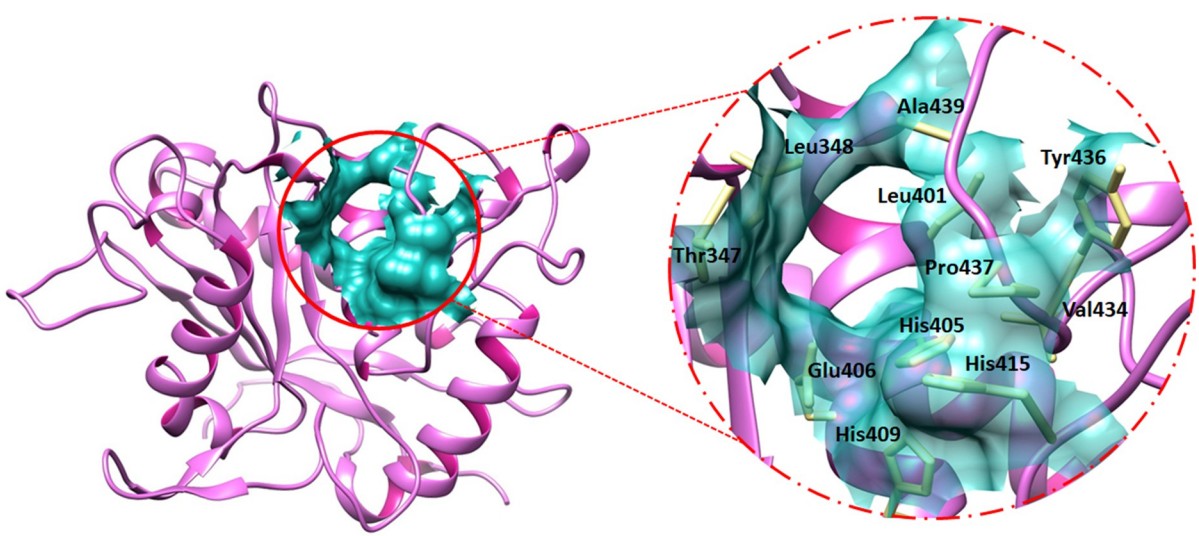

**Fig 6. Figure manifests the real-time location of the active region of TACE on the left while the active pocket amino acids are depicted on the right side of the image.**

compound BMS-561392 were ranked based on their minimum docking energy in molecular docking studies (S1 Table).

Quinapril, Vorinostat, and Alvimopan demonstrated the lowest CDocker energy (-63.0703, -54.7973, -52.7959 kcal/mol respectively) among the compounds analyzed, suggesting they have the highest binding strength in molecular docking studies. They also exhibited notably low CDocker interaction energy marking it as a highly potent candidate against TACE. Although Vorinostat and Alvimopan had a slightly higher CDocker energy than Quinapril, they showed very favorable interaction energy (-60.0783, -65.4044 kcal/mol respectively), indicating robust interactions within the binding site. Marimastat, a known TACE inhibitor, recorded a CDocker energy of -49.1011 kcal/mol, reflecting its good binding affinity, though it was slightly higher than other top contenders. BMS-561392, the reference drug, had a CDocker energy of -46.8811 kcal/mol, which was relatively higher, yet it displayed the lowest interaction energy (-84.7487 kcal/mol) among all the compounds, suggesting exceptionally favorable interactions despite its overall binding affinity being lower.

Other compounds such as Bufexamac (-44.9771 kcal/mol), Loxoprofen (-44.2524 kcal/mol), and Thiamine (-41.7703 kcal/mol) also showed competitive CDocker energies, indicating their potential as effective inhibitors, although their interaction energies varied, with Loxoprofen having the most favorable among them. The significance of these findings extends to drug development, where compounds with lower CDocker energy values are generally preferred due to their higher likelihood of effective binding and stability. Quinapril, Vorinostat, and Alvimopan with their exceptionally high negative CDocker energy, emerge as a promising candidate for further investigation.

Similarly, Marimastat, BMS-561392, Loxoprofen, Bufexamac, and Thiamine with their relatively good binding and interaction profiles, could be strong candidates for therapeutic applications. Despite its higher CDocker energy, BMS-561392's extremely low interaction energy may still render it a potent option against the target protein (Table 2).

Understanding these energy values is crucial as they help in predicting the efficacy of the compounds in a biological setting. Compounds that bind more stably are less likely to dissociate from the target, enhancing their potential as effective drugs. The data provided here serves

**Table 2.  The CDocker and CDocker interactions score of the top 15 docked compounds against TACE.**

| No | Compounds | CDocker energy (kcal/mol) | CDocker interaction energy (kcal/mol) |
|---|---|---|---|
| 1 | Quinapril | -63.0703 | -79.1213 |
| 2 | Vorinostat | -54.7973 | -60.0783 |
| 3 | Alvimopan | -52.7959 | -65.4044 |
| 4 | Marimastat | -49.1011 | -57.4456 |
| 5 | BMS-561392 (Ref) | -46.8811 | -84.7487 |
| 6 | Bufexamac | -44.9771 | -51.3456 |
| 7 | Loxoprofen | -44.2524 | -58.5632 |
| 8 | Thiamine | -41.7703 | -45.0779 |
| 9 | Adrafinil | -41.0990 | -50.5074 |
| 10 | Naproxen | -37.2369 | -53.9294 |
| 11 | Phenformin | -36.9916 | -41.0075 |
| 12 | Zinc_Acetate | -35.5984 | -33.0282 |
| 13 | Cetirizine | -34.6891 | -54.0816 |
| 14 | Liarozole | -32.0840 | -45.1788 |
| 15 | Detomidine | -31.7856 | -34.8698 |

as a foundational step in the selection and optimization of lead compounds for further analysis.

## Binding interaction analysis of screened compounds

The molecular docking results were further analyzed by examining the interactions between the compounds and the active region amino acids target protein, TACE, along with the corresponding binding distances in coordination with Zn2+. BMS-561392, the reference drug, interacted with Glu406 and Gly349, showing binding distances ranging from 1.97Å to 2.72Å. This interaction correlates with its low CDocker interaction energy, despite its relatively higher CDocker energy. Quinapril, which had the lowest CDocker energy, formed interactions with Gly346, Leu348, and Gly349, with binding distances between 2.02Å and 2.68Å, supporting its strong binding interaction. Vorinostat, which also exhibited low CDocker energy, interacted with Gly349, Glu406, and Tyr436, with binding distances from 2.06Å to 2.64Å, highlighting its robust interaction profile. Alvimopan, another strong candidate, interacted with Gly349 and Leu348, with distances of 2.68Å and 2.39Å, respectively, which aligns with its favorable interaction energy. Marimastat, with its slightly higher CDocker energy, interacted closely with Gly349 and Glu406, with binding distances of 1.91Å and 2.23Å, reflecting its known inhibitory effects. Bufexamac, Loxoprofen, and Thiamine, which had competitive CDocker energies, interacted primarily with Gly349, His405, and Glu406, respectively, with binding distances around 2.00Å to 2.27Å, indicating their potential efficacy (Fig 7). These binding interactions provide insight into the molecular basis of the docking results, where compounds with lower CDocker energies and shorter binding distances generally exhibit more stable and effective interactions, highlighting their potential as therapeutic agents (Table 3).

The Zn2+ ion plays a pivotal role in the molecular docking and binding interactions within the active site of TACE. As a metalloproteinase, TACE's enzymatic activity heavily dependent on the coordination environment of the Zn2+ ion, which serves as a critical cofactor. The Zn2+ ion typically coordinates with key amino acid residues, such as Glu406, His405, and Gly349, forming a stable, catalytically active site that is crucial for ligand binding. In molecular docking studies, the presence of Zn2+ significantly influences the binding affinity and orientation of ligands. Compounds that can effectively interact with Zn2+ or its coordinating residues often exhibit more stable and potent binding conformation. Most of the ligands formed the interaction with the neighboring amino acids of Zn2+ thus showing the potency of the screened ligands to be further evaluated.

## Molecular dynamics (MD) simulations

MD simulation reveals potential changes in binding modes, and the durability of interactions under physiological conditions, and identifies any transient states that contribute to their inhibitory activity. To assess the stability of the screened compounds against TACE, the docked complexes were subjected to 100ns MD simulations using GROMACS.

**Root mean square deviation.** The molecular dynamics trajectories were analyzed by computing the Root Mean Square Deviation (RMSD) to evaluate ligand fluctuations within the active site of the TACE protein.

BMS-561392, Vorinostat, and Bufexamac exhibited similar behavior in terms of their RMSD values, showing the lowest RMSD and highly stable profiles throughout the simulation, suggesting a strong correlation with their favorable docking scores and interaction energies. This stability indicates that these compounds maintain consistent binding within the active site, reinforcing their potential as potent inhibitors. Thiamine and Quinapril, while also displaying stable RMSD profiles, had relatively higher RMSD values compared to Vorinostat and

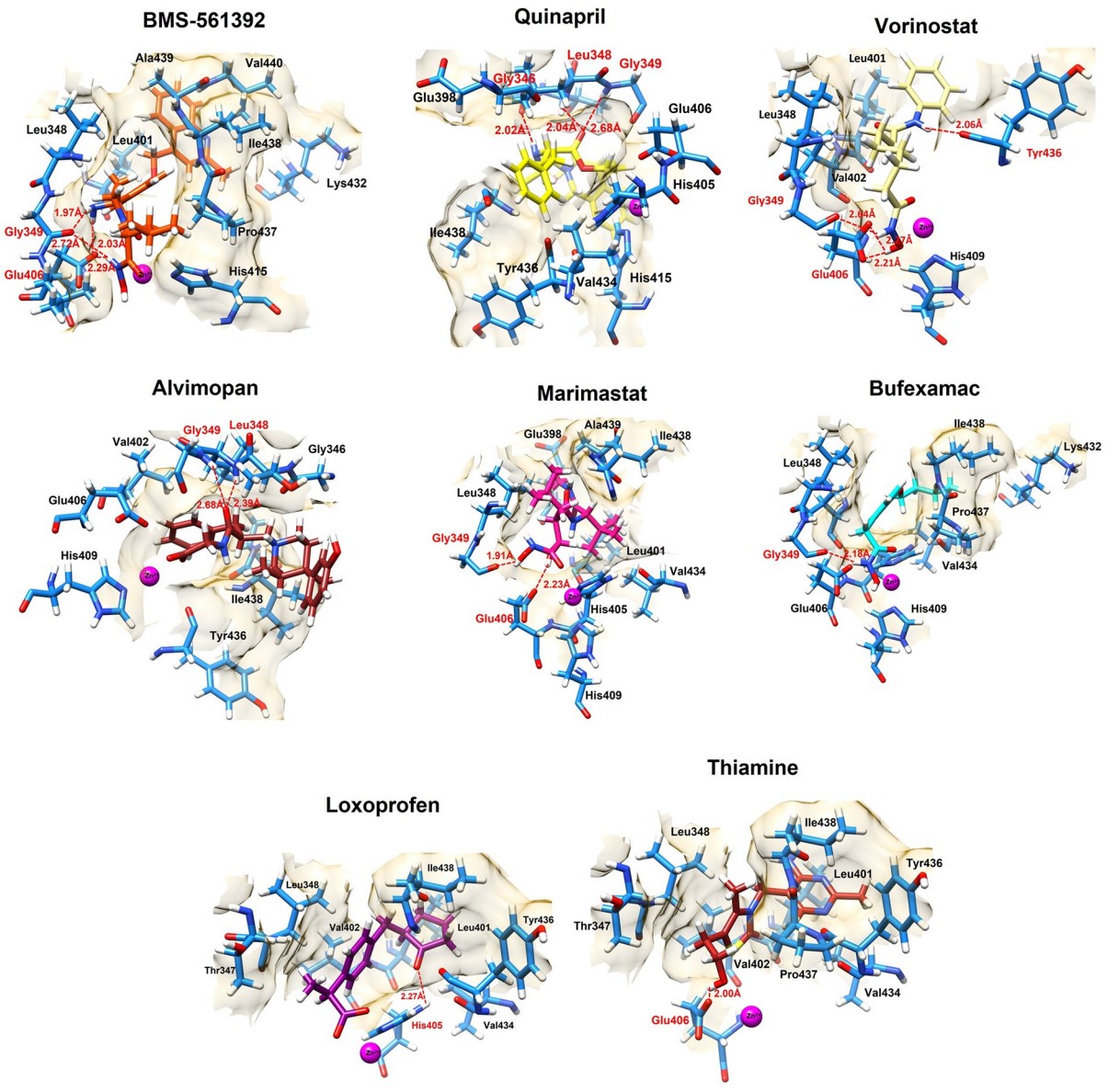

**Fig 7. The 3D interactions of the docked ligands were depicted in this figure, confirming that the ligands bound in the active region of the target protein.** The interactions, including hydrogen bonds and interacting amino acids, are depicted in red. The interacting Zn2+ ion crucial for TACE binding was colored magenta.

BMS-561392. Marimastat, despite its slightly higher CDocker energy, exhibited notable transitions in its RMSD profile. Initially, the RMSD increased, followed by a decrease at around 28ns, and then a re-increase, with a significant drop to 0.56Å at 85ns, which remained stable until the end of the simulation. This fluctuation might reflect Marimastat's binding dynamics, where the compound adjusts within the active site before stabilizing, potentially indicating a flexible binding mode that could influence its inhibitory efficacy. Alvimopan's RMSD profile revealed fluctuations from start to finish, with values oscillating between 0.3Å and 0.5Å, and a peak of 0.72Å at 8ns. This variability might be attributed to the compound's interactions within the binding site, which, while stable, involve slight positional shifts that are less stable

**Table 3. The interacting amino acids and the bond lengths are manifested in this table of the top-screened FDA compounds.**

| Drugs | Interacting amino acids | Binding distances | Common residues |
|---|---|---|---|
| BMS-561392 | Glu406<br>Gly349 | 2.03Å, 2.29Å<br>1.97Å, 2.72Å | Glu406, Gly349, Leu348 |
| Quinapril | Gly346<br>Leu348<br>Gly349 | 2.02Å<br>2.04Å<br>2.68Å | |
| Vorinostat | Gly349<br>Glu406<br>Tyr436 | 2.64Å<br>2.27Å, 2.21Å<br>2.06Å | |
| Alvimopan | Gly349<br>Leu348 | 2.68Å<br>2.39Å | |
| Marimastat | Gly349<br>Glu406 | 1.91Å<br>2.23Å | |
| Bufexamac | Gly349 | 2.18Å | |
| Loxoprofen | His405 | 2.27Å | |
| Thiamine | Glu406 | 2.00Å | |

than those seen in compounds like Vorinostat or BMS-561392. Loxoprofen, on the other hand, showed a relatively stable RMSD compared to Alvimopan and Marimastat, with only one significant peak between 39ns and 60ns before re-stabilizing. This suggests that while Loxoprofen's binding is generally stable, it may undergo transient conformational changes before achieving equilibrium (Fig 8).

The observed RMSD behaviors, when compared with the docking scores, suggest that compounds with lower RMSD values tend to have more stable and consistent binding, aligning with their lower CDocker energies. The stable RMSD values of Vorinostat and bufexamac revealed a strong interaction while fluctuations in RMSD for Marimastat and Alvimopan could indicate a more dynamic interaction within the binding site. These results imply that while stable RMSD profiles correlate with strong binding affinity, the flexibility seen in some compounds may reflect the potential for adaptive binding mechanisms that could be further explored for therapeutic applications.

In addition to analyzing the ligand-protein RMSD, backbone-to-backbone RMSD calculations were also conducted to assess whether the proteins reached equilibrium during the

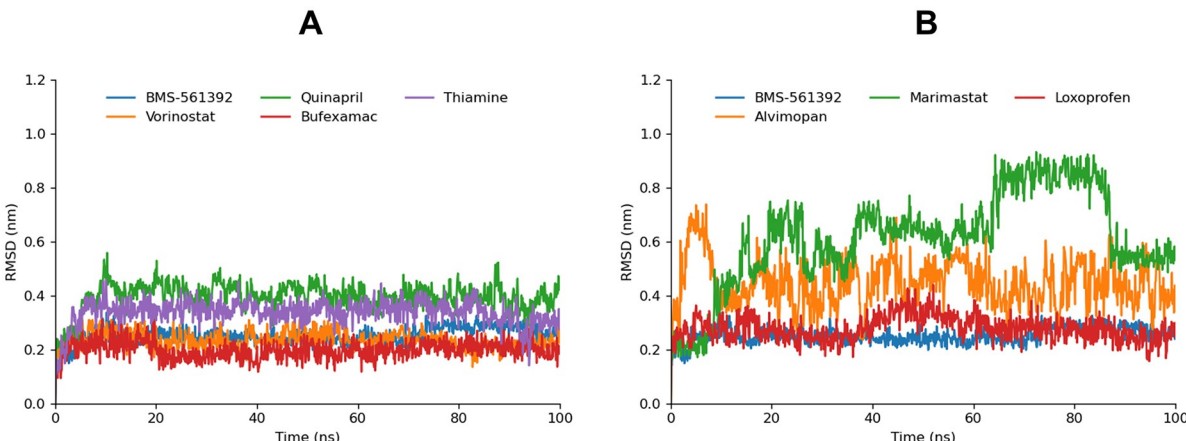

**Fig 8. The graphs illustrate the RMSD plot of the ligands.** The simulated ligands were colored differently for comparison. The less fluctuating bar graphs are depicted in graph Fig A and compounds with moderate to high fluctuations are manifested in Fig B.

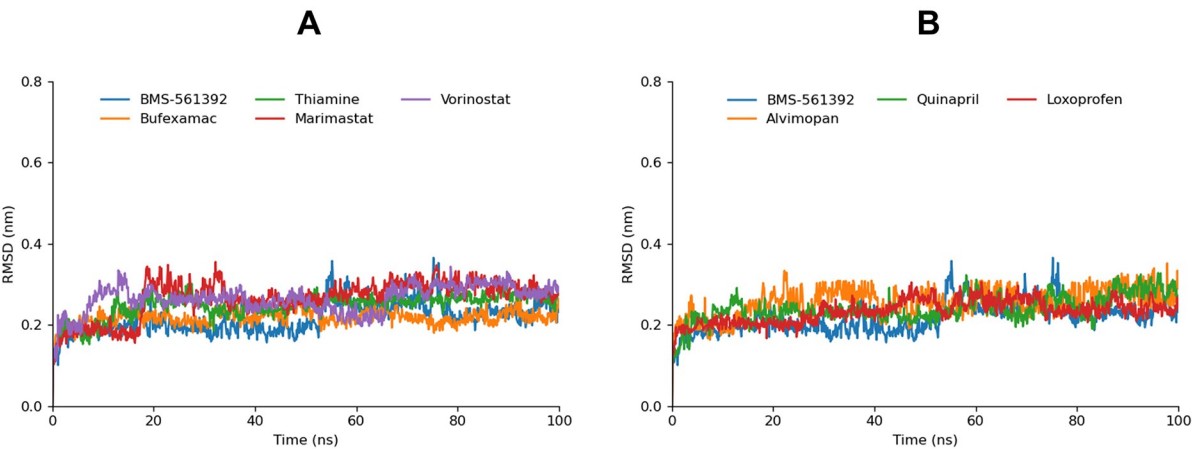

**Fig 9. The graphs illustrate the RMSD plots of the protein backbone.** Graph A manifests the RMSD backbone of Thiamine, Vorinostat, Bufexamac, and Marimastat compared to BMS-561392. Graph B exhibits the RMSD backbone of Quinapril, Alvimopan, and Loxoprofen compared to BMS-561392.

simulation (Fig 9). The resulting bar graphs revealed a consistent and stable behavior across the board, indicating that the overall protein structures maintained their integrity and did not undergo significant conformational changes. This stability in the backbone RMSD suggests that the proteins were well equilibrated throughout the simulation, providing a reliable framework for analyzing the ligand interactions.

**Hydrogen bond plot analysis.** The hydrogen bond plot analysis distinguishes between two types of hydrogen bonds: actual hydrogen bonds, which were identified during the 100ns molecular dynamics (MD) trajectory, and potential hydrogen bonds, where the atoms of the ligands and receptor are close enough (within 0.35 nm) to possibly form hydrogen bonds in future trajectories (Fig 10).

The hydrogen bond plot analysis revealed significant insights into the binding dynamics of various compounds with the target protein, TACE. BMS-561392, the reference compound, consistently formed two actual and two potential hydrogen bonds, underscoring its strong and stable interaction with the target. Vorinostat also demonstrated robust binding throughout the 100 ns MD trajectory, forming three actual hydrogen bonds and one potential hydrogen bond, along with numerous additional peaks for potential hydrogen bonds, indicating its strong candidacy as a TACE inhibitor. Bufexamac showed stable interactions with two actual and one potential hydrogen bond, the peaks of 4th and 5th potential hydrogen bonds can also be seen. Marimastat also had two actual hydrogen bonds in the initial trajectory while after 80ns the number of actual hydrogen bonds were reduced but the number of potential hydrogen bonds remain consistent.

In comparison, Quinapril, Thiamine and Loxoprofen maintained one actual hydrogen bond, with the second actual and potential hydrogen bonds alternating, indicating intermittent interactions. Alvimopan on the initial trajectory manifested low interactions but after 55ns the number of interactions increased followed by the decrease at the end, corelating the RMSD. Quinapril, Thiamine and Loxoprofen showed weaker interactions overall, with Alvimopan notably increasing its potential hydrogen bonds in the second half of the simulation, suggesting a delayed or less binding effect. These findings align with the RMSD results, providing a comprehensive understanding of the binding dynamics and potential efficacy of these

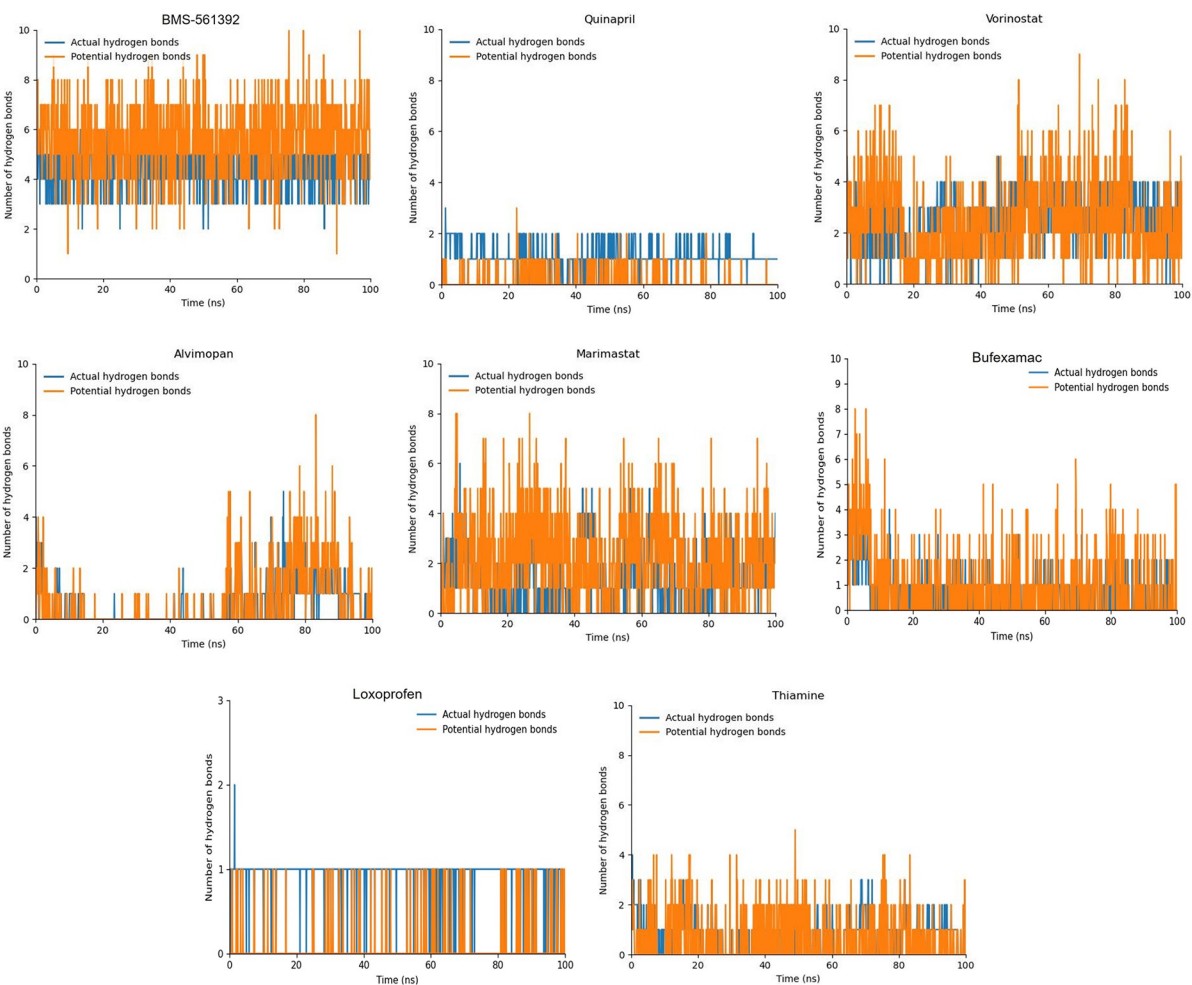

**Fig 10. The graphs show the hydrogen bond plots of the top 10 docked compounds against TACE.** The blue peaks are the actual hydrogen bonds while the orange peaks are potential hydrogen bonds.

compounds. Further studies are necessary to validate these observations and explore their therapeutic potential as TACE inhibitors in clinical settings.

**Interaction energy analysis.** In addition to hydrogen bond visualization and RMSD analysis, the interaction energies of the top 10 docked compounds were calculated against TACE during a 100ns MD simulation to evaluate the interaction energy scores of the docked complexes. The interaction energy consists of two components: electrostatic (Coulombic) interaction energy and Lennard-Jones interaction energy, with their sum representing the total interaction energy.

BMS-561392 as a reference drug exhibited the lowest total interaction energy (-103.1152 kcal/mol). Among the analyzed compounds, Vorinostat (-51.5201 kcal/mol) indicated the strongest binding affinity, followed by Thiamine (-49.0065 kcal/mol). Quinapril, which had the lowest CDocker energy score, with a total energy of -42.7971 kcal/mol, also showed significant binding strength. In comparison, Loxoprofen had the highest total interaction energy (-0.6898 kcal/mol), suggesting weaker binding (Table 4). These interaction energy results correlate with RMSD and hydrogen bond analyses, which provide additional insights into the binding stability and specificity of these compounds. Vorinostat's strong binding affinity and

**Table 4. The Electrostatic, Lennard-Jones contributions and the total energies are shown.**

| No | Compound | Interaction energy (Kcal/mol) | | |
|---|---|---|---|---|
| | | Coul-SR | LJ-SR | Total energy |
| | BMS-561392(Ref) | -51.1147 | -52.0005 | -103.1152 |
| 1 | Vorinostat | -16.6317 | -34.8884 | -51.5201 |
| 2 | Thiamine | -24.2450 | -24.7615 | -49.0065 |
| 3 | Quinapril | -13.1931 | -29.6040 | -42.7971 |
| 4 | Marimastat | -13.7749 | -28.0409 | -41.8158 |
| 5 | Bufexamac | -9.33100 | -24.9137 | -34.2447 |
| 6 | Alvimopan | -11.8919 | -18.6340 | -30.5259 |
| 7 | Loxoprofen | 30.3566 | -31.0464 | -0.6898 |

comprehensive interaction profile, as reflected in both its low total interaction energy and favorable RMSD and hydrogen bond metrics, underscore its potential as a highly potent therapeutic agent. Bufexamac, while also displaying low interaction energies, may have slightly less potency compared to Vorinostat due to their fewer binding interactions as indicated by RMSD and hydrogen bond analyses. Overall, this integrated approach combining interaction energy data with structural and interaction analyses provides a robust framework for evaluating the efficacy of these compounds in drug development.

The sum of Coulombic and Lennard-Jones interaction energy (Total energy) of the screened FDA drugs was graphically depicted in Fig 11 in comparison with BMS-561392.

## gmx_MMPBSA Binding free energy calculation

The entire trajectories obtained during 100ns MD simulation was divided into five groups, therefore was subjected to calculate the binding free energy of docked complexes. The gmx_MMPBSA tool used with GROMACS was employed and the MM/PBSA method was used to calculate the binding energy with default parameters (Fig 12A and 12B).

The average -ΔG values along with their Standard Deviation (SD) for screened compounds evaluated for their binding affinity to a target protein in comparison with the reference compound. Notably, Loxoprofen revealed very low binding affinity therefore removed from the

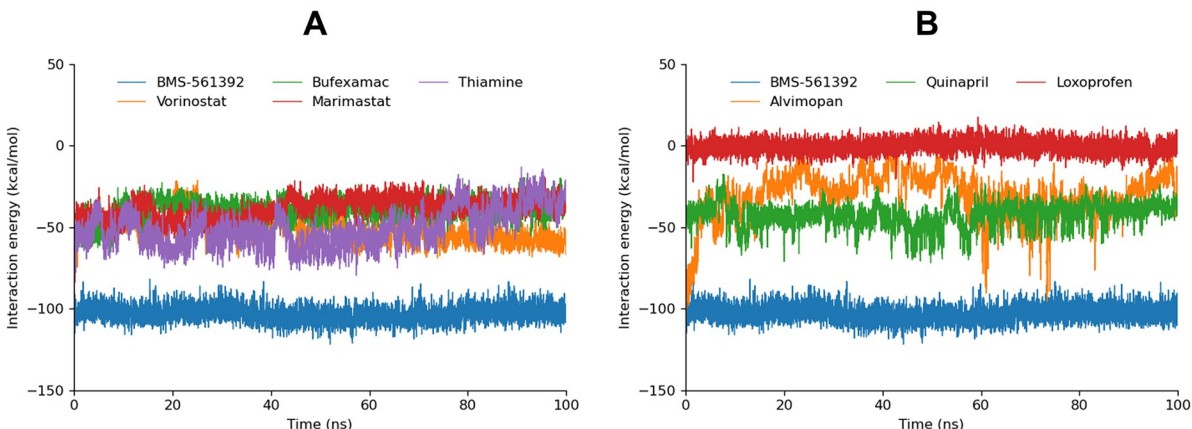

**Fig 11. The computed total interaction energies of Enamine compounds, derived by summing the Lennard-Jones and Coulombic contributions, are presented here in comparison with BMS-561392.**

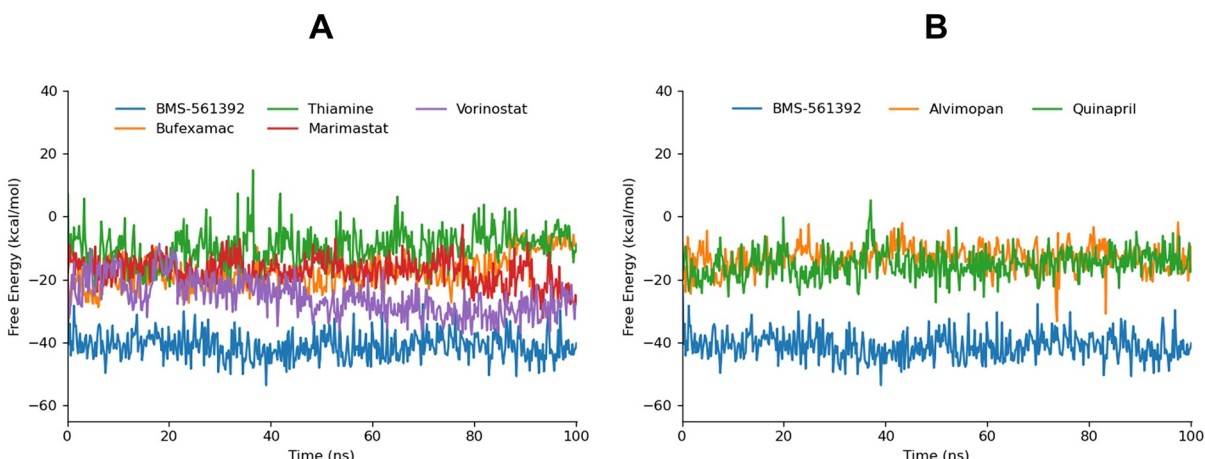

**Fig 12. The graphical depiction of gmx_MMPBSA free energy calculation of top 6 compounds in comparison with BMS-561392.**

data set. BMS-561392 exhibited the most favorable binding energy with ΔG of -40.72 kcal/mol, indicating strong binding affinity. This is complemented by a low average SD value of 3.54, suggesting consistency in the calculated energy values across simulations. Vorinostat also demonstrated significant binding affinity with ΔG -25.80 kcal/mol with a SD value of 3.67, indicating robustness in its binding interactions. Bufexamac showed ΔG values of -18.37 kcal/mol with moderate SD (3.42), suggesting stable but less favorable binding compared to BMS-561392 and Vorinostat. Furthermore, Marimastat and Quinapril displayed ΔG values of -17.41 kcal/mol and -15.04 kcal/mol with relatively lower SD values, indicating consistent but weaker binding affinities compared to BMS-561392 and Vorinostat (Table 5).

The ΔG values correlate with the interaction energies observed in the hydrogen bond plot analysis. Compounds with lower -ΔG, such as BMS-561392 and Vorinostat, generally exhibited more stable hydrogen bond interactions with the target protein over the 100 ns MD simulation. Conversely, compounds like Thiamine and Alvimopan, which showed higher -ΔG values of -8.97 kcal/mol and -13.34 kcal/mol respectively, also displayed less favorable and less stable hydrogen bond patterns. Overall, these findings highlight Vorinostat as a promising candidate for further investigation as a TACE inhibitor, supported by its strong -ΔG values and stable interaction profiles followed by Bufexamac. Further experimental validation is crucial to confirm these computational predictions and evaluate their potential therapeutic efficacy.

**Table 5. The binding affinity of the top 10 simulated compounds in comparison with BMS-561392.**

| Compound | 1ns-20ns | | 20ns-40ns | | 40ns-60ns | | 60ns-80ns | | 80ns-100ns | | Average | |
|---|---|---|---|---|---|---|---|---|---|---|---|---|
| | ΔG | SD | ΔG | SD | ΔG | SD | ΔG | SD | ΔG | SD | ΔG | SD |
| BMS-561392 | -39.01 | 5.09 | -41.43 | 2.96 | -41.90 | 4.58 | -39.77 | 2.36 | -41.51 | 2.71 | -40.72 | 3.54 |
| Vorinostat | -20.45 | 5.68 | -21.41 | 3.77 | -28.39 | 3.12 | -29.40 | 2.03 | -29.33 | 3.74 | -25.80 | 3.67 |
| Bufexamac | -18.70 | 4.91 | -20.09 | 3.55 | -19.69 | 2.61 | -17.01 | 3.41 | -16.39 | 2.60 | -18.37 | 3.42 |
| Marimastat | -15.04 | 3.73 | -16.86 | 4.18 | -16.51 | 2.61 | -16.28 | 3.29 | -22.38 | 4.29 | -17.41 | 3.62 |
| Quinapril | -16.56 | 3.23 | -15.25 | 5.10 | -15.65 | 3.98 | -15.00 | 3.56 | -12.72 | 3.01 | -15.04 | 3.77 |
| Alvimopan | -15.17 | 3.80 | -13.33 | 2.50 | -11.69 | 2.64 | -14.06 | 5.04 | -12.49 | 3.70 | -13.34 | 3.53 |
| Thiamine | -12.21 | 3.35 | -8.41 | 5.03 | -8.88 | 5.27 | -8.370 | 4.94 | -7.00 | 3.79 | -8.97 | 4.48 |

**Table 6. Tanimoto similarity comparison of selected compounds.**

| Names | BMS-561392 | Marimastat | Vorinostat | Bufexamac | Thiamine | Alvimopan | Quinapril |
|---|---|---|---|---|---|---|---|
| BMS-561392 | - | 0.313 | 0.123 | 0.205 | 0.280 | 0.348 | 0.444 |
| Marimastat | 0.313 | - | 0.172 | 0.123 | 0.181 | 0.268 | 0.283 |
| Vorinostat | 0.123 | 0.172 | - | 0.150 | 0.096 | 0.124 | 0.102 |
| Bufexamac | 0.205 | 0.123 | 0.150 | - | 0.127 | 0.210 | 0.169 |
| Thiamine | 0.280 | 0.181 | 0.096 | 0.127 | - | 0.187 | 0.235 |
| Alvimopan | 0.348 | 0.268 | 0.124 | 0.210 | 0.187 | - | 0.311 |
| Quinapril | 0.444 | 0.283 | 0.102 | 0.169 | 0.235 | 0.311 | - |

## Structural evaluation and similarity comparison

To evaluate the structural similarities among the selected compounds for experimental assessment, we utilized RDKit's Tanimoto similarity measure. The analysis indicated that there was no significant structural resemblance between the selected compounds and BMS-561392, as shown by the Tanimoto similarity (Table 6). While there is no strict threshold for defining similarity, a Tanimoto similarity score of 0.8 or higher generally suggests similarity, ranging from 0 (no similarity) to 1 (complete similarity). Although the selected compounds exhibited limited overall similarity to BMS-561392, they may still share specific common structural motifs. To identify these common substructures, we employed the Maximum Common Substructure (MCS) algorithm in RDKit, which utilizes SMARTS (SMILES Arbitrary Target Specification). No recurring structural motifs were identified, aside from the benzene ring highlighted in green (Fig 13). This observation suggests that factors beyond structural motifs, such as the spatial orientation of specific conformations, could potentially influence the

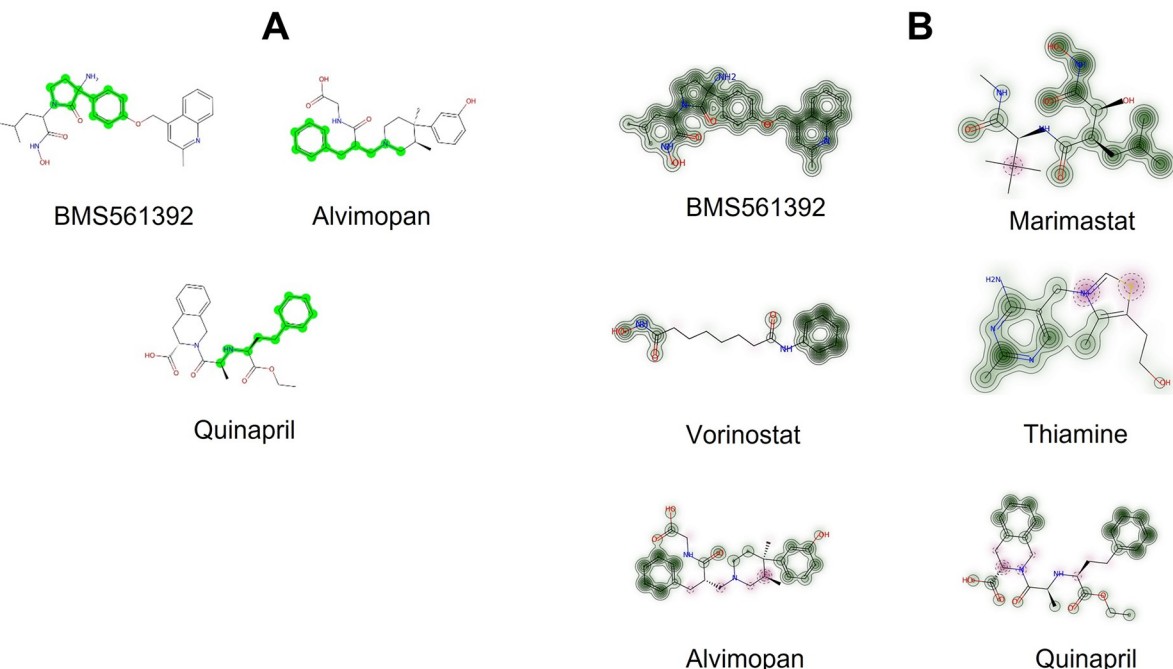

**Fig 13. Visual depiction of the common structural motif identified through Maximum Common Substructure (MCS) analysis using SMARTS (A), and graphical representation illustrating the shared common structural motif through similarity maps (B).**

inhibitory activity against TACE. Additionally, we utilized RDKit fingerprints to generate similarity maps to determine if the selected compounds contain structural motifs similar to those in BMS-561392 (Fig 13). The similarity maps indicated the presence of BMS-561392 structural motifs within the chemical structures of the selected compounds. These insights from the MCS and similarity mapping provide valuable guidance for further optimizing the selected compounds.

Overall, the computational analysis of potential TACE inhibitors provided valuable insights into the binding affinities and molecular interactions of the selected compounds. Vorinostat, identified as the most promising candidate, demonstrated a significantly highly negative binding free energy and consistent interaction with key residues in the TACE active site. These findings align with prior research highlighting the potential of hydroxamate-containing compounds as effective TACE inhibitors due to their ability to chelate the catalytic $Zn^{2+}$ ion within the active site [9,22]. Such interactions were further validated through our molecular dynamics (MD) simulations, which confirmed the stability of the these complexes over a 100 ns trajectory. Recent advances in computational modeling and dynamics simulations have significantly enhanced our understanding of drug design [44]. Similarly, leveraging tools such as GROMACS with detailed parameterization of force fields has proven effective in replicating realistic biomolecular interactions, enabling high-fidelity simulation of protein-ligand complexes [45–50].

The inclusion of entropy contributions to the free energy calculations allowed for a comprehensive evaluation of the binding thermodynamics, emphasizing the importance of entropic factors in ligand binding. The relatively lower standard deviation observed in the recalculated ΔG values enhances the reliability of these predictions, corroborating the robustness of the gmx_MMPBSA methodology employed [51–53]. BMS-561392 the reference compound, although has been extensively studied, its clinical use has been limited due to toxicity concerns, making the identification of safer alternatives critical [54,55].

Our findings also highlight the dynamic interaction with the key residues such as Glu406, His405, and Tyr436 in stabilizing ligand binding through hydrogen bonding and hydrophobic interactions. This is consistent with previously reported binding pocket residues [28]. Moreover, our analysis of FDA-approved drug candidates underscores the potential of repurposing approved drugs for TACE inhibition, a strategy supported by prior successful case studies in drug development [56–58]. By employing an integrated computational workflow that included graph convolutional models for initial screening, docking, MD simulations, and MMPBSA calculations, this study demonstrates a reliable and reproducible approach for virtual screening and prioritization of TACE inhibitors. Therefore, encourages the experimental validations of these screened compounds.

## Experimental validation

To validate the computational findings, the TACE inhibitory activity of six compounds was assessed by measuring the release level of TNF-α at a concentration of 10 μM in RAW 263.7 macrophage cells. BMS-561392, used as the reference compound, showed the lowest TNF-α release, indicating the strongest inhibition of TACE. Marimastat, another known TACE inhibitor that was also highly predicted by the GraphConvMol algorithm in the present study, demonstrated significant suppression of TACE. Notably, Vorinostat exhibited potent inhibitory activity on TACE (Fig 14). Vorinostat is an anticancer medication used to treat cutaneous T-cell lymphoma by inhibiting histone deacetylase (HDAC) [59–61]. Prior to this study, there were no reports of Vorinostat affecting TACE activity. In contrast, Thiamine showed the least

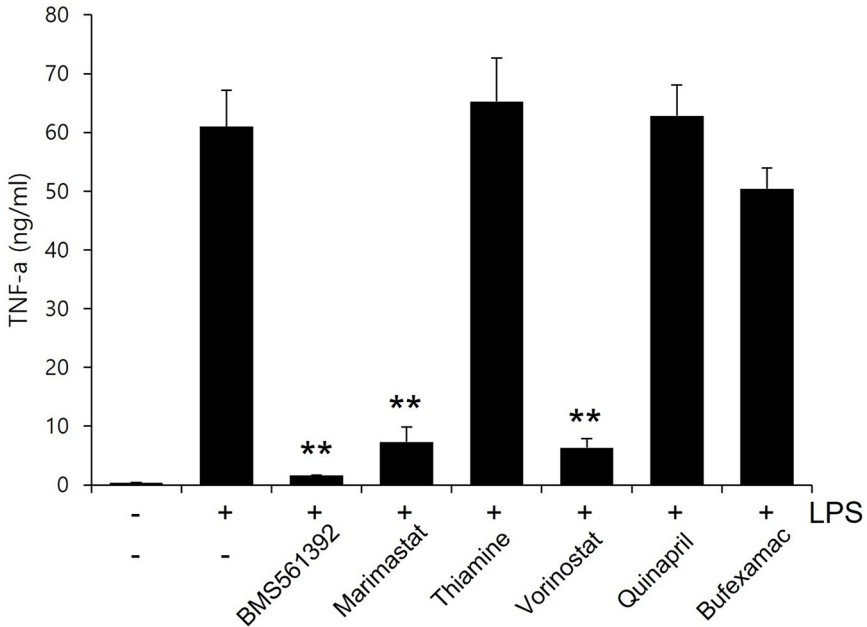

**Fig 14. TACE inhibitory activity of selected compounds in comparison to BMS-561392.**

effect, suggesting weaker inhibition. Other compounds, including Bufexamac and Quinapril displayed moderate to high enzymatic activities, reflecting a range of inhibitory potential.

The enzymatic activity results align well with the gmx_MMPBSA and MD simulation outcomes. BMS-561392, which showed the lowest enzymatic activity, also had the most favorable binding energies and stability in MD simulations, supporting its strong inhibitory effect on TACE. Vorinostat, which demonstrated significant inhibition in enzymatic assays, was predicted to have strong binding affinity and stability in the gmx_MMPBSA and MD simulations, reinforcing its potential as a TACE inhibitor. Marimastat's moderate enzymatic activity corresponds with its consistent performance in computational analyses, validating its known efficacy. Conversely, Thiamine's low enzymatic activity, indicating weak inhibition, is consistent with its lower binding affinity in the gmx_MMPBSA. Other compounds like Quinapril and Bufexamac showed moderate binding affinities and stabilities in MD simulations, which correlate with their enzymatic activity results, highlighting their varied inhibitory potentials. This comprehensive analysis demonstrates the reliability of integrating computational and experimental approaches.

## Conclusion

In conclusion, this study demonstrates the powerful application of deep learning models in the drug discovery process, particularly in identifying novel inhibitors for specific targets such as TACE. By utilizing advanced cheminformatics tools and a robust predictive framework, we successfully screened a library of FDA-approved drugs and identified several promising TACE inhibitors, including Vorinostat, Quinapril, Alvimopan, and Marimastat. Vorinostat, previously known as an anticancer medication that works by inhibiting histone deacetylase, has not been reported to inhibit TACE prior to our findings. Our computational predictions, supported by molecular docking experiments, confirmed Vorinostat's strong binding affinity and inhibitory activity against TACE, suggesting its potential for repurposing in the treatment of

inflammatory diseases like rheumatoid arthritis. Similarly, Quinapril demonstrated the lowest docking energy, indicative of its strong potential to bind and inhibit TACE effectively. Alvimopan and Marimastat also exhibited favorable binding properties, with molecular docking and dynamic simulation studies revealing stable interactions with critical residues in the TACE active site.

These findings underscore the efficiency and effectiveness of AI-driven approaches in accelerating drug repurposing and discovery efforts. The identification of potential TACE inhibitor, such as Vorinostat, opens new avenues for therapeutic intervention and highlights the versatility of this compound beyond its original indications. This drugs demonstrated unique interaction profiles with the TACE active site suggesting a diverse mechanism of inhibition that warrants further experimental validation and optimization.

## Supporting information

**S1 Table. Docking energy score of all 33 docked compounds.**
(DOCX)

## Author Contributions

**Conceptualization:** Wanjoo Chun.

**Data curation:** Eun-Taek Han, Jin-Hee Han, Won Sun Park.

**Formal analysis:** Muhammad Yasir, Jinyoung Park.

**Investigation:** Jinyoung Park.

**Writing – original draft:** Muhammad Yasir, Wanjoo Chun.

**Writing – review & editing:** Mubashir Hassan, Andrzej Kloczkowski, Wanjoo Chun.

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
