## [Decision Letter · Decision Letter 0]

13 Nov 2024

PONE-D-24-43186Discovery of Novel TACE Inhibitors Using Graph Convolutional Network, Molecular Docking, Molecular Dynamics Simulation, and Biological EvaluationPLOS ONE

Dear Dr. chun,

Thank you for submitting your manuscript to PLOS ONE. After careful consideration, we feel that it has merit but does not fully meet PLOS ONE’s publication criteria as it currently stands. Therefore, we invite you to submit a revised version of the manuscript that addresses the points raised during the review process. Please submit your revised manuscript by Dec 28 2024 11:59PM. If you will need more time than this to complete your revisions, please reply to this message or contact the journal office at plosone@plos.org. Please include the following items when submitting your revised manuscript:A rebuttal letter that responds to each point raised by the academic editor and reviewer(s). You should upload this letter as a separate file labeled 'Response to Reviewers'.A marked-up copy of your manuscript that highlights changes made to the original version. You should upload this as a separate file labeled 'Revised Manuscript with Track Changes'.An unmarked version of your revised paper without tracked changes. You should upload this as a separate file labeled 'Manuscript'.If applicable, we recommend that you deposit your laboratory protocols in protocols.io to enhance the reproducibility of your results. Protocols.io assigns your protocol its own identifier (DOI) so that it can be cited independently in the future. For instructions see: https://journals.plos.org/plosone/s/submission-guidelines#loc-laboratory-protocols. Additionally, PLOS ONE offers an option for publishing peer-reviewed Lab Protocol articles, which describe protocols hosted on protocols.io. Read more information on sharing protocols at https://plos.org/protocols?utm_medium=editorial-email&utm_source=authorletters&utm_campaign=protocols.

We look forward to receiving your revised manuscript.

Kind regards,

Cheorl-Ho Kim, Ph.D.

Academic Editor

PLOS ONE

Journal Requirements:

3. Thank you for stating the following financial disclosure: [This work was supported by a Research Grant from the Institute of Medical Sciences,

Kangwon National University 2024, a Korea Basic Science Institute (National Research Facilities and Equipment Center) grant funded by the Ministry of Education (grant no. 2022R1A6C101A739), and NIH grant numbers R01GM127701 and R01HG012117.]. Please state what role the funders took in the study. If the funders had no role, please state: "The funders had no role in study design, data collection and analysis, decision to publish, or preparation of the manuscript." If this statement is not correct you must amend it as needed. Please include this amended Role of Funder statement in your cover letter; we will change the online submission form on your behalf.

4. Please expand the acronym “NIH” (as indicated in your financial disclosure) so that it states the name of your funders in full. This information should be included in your cover letter; we will change the online submission form on your behalf.

5. Thank you for stating the following in the Acknowledgments Section of your manuscript: [WC acknowledges the Research Grant from the Institute of Medical Sciences, Kangwon National University 2024 and a Korea Basic Science Institute (National Research Facilities and Equipment Center) grant funded by the Ministry of Education (grant no. 2022R1A6C101A739). A.K. acknowledges financial support from NIH Grants R01 GM127701 and R01HG012117. M.H. acknowledges The Ohio State University for the “President’s Postdoctoral Scholars Program (PPSP)” fellowship for financial support.] We note that you have provided funding information that is not currently declared in your Funding Statement. However, funding information should not appear in the Acknowledgments section or other areas of your manuscript. We will only publish funding information present in the Funding Statement section of the online submission form. Please remove any funding-related text from the manuscript and let us know how you would like to update your Funding Statement. Currently, your Funding Statement reads as follows: [This work was supported by a Research Grant from the Institute of Medical Sciences,

Kangwon National University 2024, a Korea Basic Science Institute (National Research Facilities and Equipment Center) grant funded by the Ministry of Education (grant no. 2022R1A6C101A739), and NIH grant numbers R01GM127701 and R01HG012117.] Please include your amended statements within your cover letter; we will change the online submission form on your behalf.

Additional Editor Comments:

Thank you for your kind submission of your work and I am very pleased to inform you that your submission would be considered for publication.

Please contact me when you have any problem in revision.

Thanks a lot

Cheorl-Ho Kim

Prof

SKKU Biological Science

Suwon Korea

Reviewers' comments:

Reviewer's Responses to Questions

**Comments to the Author**

1. Is the manuscript technically sound, and do the data support the conclusions?

Reviewer #1: Yes

Reviewer #2: Yes

2. Has the statistical analysis been performed appropriately and rigorously? 

Reviewer #1: Yes

Reviewer #2: Yes

3. Have the authors made all data underlying the findings in their manuscript fully available?

Reviewer #1: Yes

Reviewer #2: Yes

4. Is the manuscript presented in an intelligible fashion and written in standard English?

Reviewer #1: Yes

Reviewer #2: Yes

5. Review Comments to the Author

Reviewer #1: The manuscript entitled “Discovery of Novel TACE Inhibitors Using Graph Convolutional Network, Molecular Docking, Molecular Dynamics Simulation, and Biological Evaluation” has many mistakes, authors need to rectify many portions.

• Line 97: The sentence "Therefore, its potential to identify novel TACE inhibitors with greater accuracy and efficiency was examined in this study" is unclear. The sentence does not specify what "its" refers to. It should be rewritten for clarity.

• Line 79: The phrase "TNF-α, is also produced by neurons and glia," includes an unnecessary comma after "TNF-α."

• Line 191: The sentence "TACE, Ligand, and Zn2+ ion was uploaded to the server for each complex and CHARMM36 force field [32] setup was performed using the solution builder protocol on the CHAR..." ends abruptly and seems incomplete.

• The author required to update recent references can be seen PMID: 37065061, 39065802, 35424125, 34297427, 35014595, 36936534.

• Check the sentence: "metrics such as the Area Under the Curve (AUC) of the Receiver Operating Characteristic (ROC) curve..."

• Confusing sentence please correct it: TPR of 1 usually indicates perfect performance at a specific threshold, but it doesn’t make sense in relation to the ROC curve unless further clarification is provided.

• It's unclear whether "perfect predictions" refers to all metrics being 1.0 or something else. A more precise explanation would enhance clarity.

• "2OI0 and co-crystallized ligand was utilized to figure out the interacting amino acid residues..."

• Check the sentence "Quinapril, Vorinostat, and Alvimopan with their exceptionally low CDocker energy, emerge as a promising candidate..."

Good Luck!

Reviewer #2: The authors work on Discovery of Novel TACE Inhibitors Using Graph Convolutional Network, Molecular Docking, Molecular Dynamics Simulation, and Biological Evaluation contributes to academic knowledge, but the discussion need to be improved upon. Further comments can be found on the annotated report

6. PLOS authors have the option to publish the peer review history of their article (what does this mean?). If published, this will include your full peer review and any attached files.

Reviewer #1: **Yes: **Shahzaib Ahamad

Reviewer #2: No

---

## [Author Response · Author response to Decision Letter 0]

20 Nov 2024

Response to Reviewers’ comments

Reviewer #1: 

The manuscript entitled “Discovery of Novel TACE Inhibitors Using Graph Convolutional Network, Molecular Docking, Molecular Dynamics Simulation, and Biological Evaluation” has many mistakes, authors need to rectify many portions.

• Line 97: The sentence "Therefore, its potential to identify novel TACE inhibitors with greater accuracy and efficiency was examined in this study" is unclear. The sentence does not specify what "its" refers to. It should be rewritten for clarity.

The suggested line has been rewritten for clarity and readability.

• Line 79: The phrase "TNF-α, is also produced by neurons and glia," includes an unnecessary comma after "TNF-α."

The Typo has been corrected.

• Line 191: The sentence "TACE, Ligand, and Zn2+ ion was uploaded to the server for each complex and CHARMM36 force field [32] setup was performed using the solution builder protocol on the CHAR..." ends abruptly and seems incomplete.

The line has been rephrased for better understanding.

• The author required to update recent references can be seen PMID: 37065061, 39065802, 35424125, 34297427, 35014595, 36936534.

All the suggested citations have been cited in the manuscript.

• Check the sentence: "metrics such as the Area Under the Curve (AUC) of the Receiver Operating Characteristic (ROC) curve..."

The sentence has been rephrased.

• Confusing sentence please correct it: TPR of 1 usually indicates perfect performance at a specific threshold, but it doesn’t make sense in relation to the ROC curve unless further clarification is provided.

The suggested sentences have been rewritten for clarity and readability.

• It's unclear whether "perfect predictions" refers to all metrics being 1.0 or something else. A more precise explanation would enhance clarity.

A precise elaboration has been provided for the clarity.

• "2OI0 and co-crystallized ligand was utilized to figure out the interacting amino acid residues..."

The sentence has been rewritten for clarity.

• Check the sentence "Quinapril, Vorinostat, and Alvimopan with their exceptionally low CDocker energy, emerge as a promising candidate..."

The sentence has been rewritten for clarity and readability.

We would like to thank you again for your careful reading and critical comments on our manuscript. We revised the manuscript based on your comments. We appreciate that your comments and suggestions significantly enhanced the quality of our work.

Reviewer #2: 

The authors work on Discovery of Novel TACE Inhibitors Using Graph Convolutional Network, Molecular Docking, Molecular Dynamics Simulation, and Biological Evaluation contributes to academic knowledge, but the discussion need to be improved upon. Further comments can be found on the annotated report

Line 108: The accession date has been updated

Line 118: The line has been rephrased according to the suggestion

Line 120: The typo mistake has been corrected. 

Line 130: The deep learning architecture from our previous publications and other publications has been discussed and cited in the manuscript.

Line 140: The lines has been rephrased

 Line 144: The citation from our previous study has been cited.

Line 172: TACE is Metalloproteinase which requires Zn2+ at the site of binding for stable interaction. The information is updated.

Line 268: We appreciate the reviewer’s insightful comment regarding the significance of the observed distribution of molecular descriptor values between active and decoy compounds. To address this concern, we conducted additional statistical analysis, calculating the mean ± standard deviation for the descriptors in both groups. The results confirm that there are no significant differences between the active and decoy groups for the evaluated descriptors. This observation is visually supported by the attached figure, which clearly illustrates the comparable distribution of values (Please see the attached figure below). 

Line 322: The Typo has been corrected

Line 590: A brief discussion has been added in comparison with previous studies validating the computational methodologies.

We would like to thank you again for your careful reading and critical comments on our manuscript. We revised the manuscript based on your comments. We appreciate that your comments and suggestions significantly enhanced the quality of our work.

---

## [Editor Report · Decision Letter 1]

22 Nov 2024

Discovery of Novel TACE Inhibitors Using Graph Convolutional Network, Molecular Docking, Molecular Dynamics Simulation, and Biological Evaluation

PONE-D-24-43186R1

Dear Dr. Chun,

We’re pleased to inform you that your manuscript has been judged scientifically suitable for publication and will be formally accepted for publication once it meets all outstanding technical requirements.

Kind regards,

Cheorl-Ho Kim, Ph.D.

Academic Editor

PLOS ONE

Additional Editor Comments (optional):

Dear Dr WJ Chun

Thank you for your submission of your study and revision.

I have checked and examined the revision and found that the revision is appropriately done.

As I have decided MINOR REVISION in a previous decision letter, I will accept your revision at this occasion.

Thank you

Sincerely

Cheorl-Ho Kim PhD Professor

SKKU

Biological Science Dept

Molecular Glycobiology

Reviewers' comments:

No more comments.

---

## [Editor Report · Acceptance letter]

12 Dec 2024

PONE-D-24-43186R1 

PLOS ONE

Dear Dr. chun, 

I'm pleased to inform you that your manuscript has been deemed suitable for publication in PLOS ONE. Congratulations! Your manuscript is now being handed over to our production team.

Kind regards, 

on behalf of

Professor Cheorl-Ho Kim 

Academic Editor

PLOS ONE